# Automated Classification of Changes of Direction in Soccer Using Inertial Measurement Units

**DOI:** 10.3390/s21144625

**Published:** 2021-07-06

**Authors:** Brian Reilly, Oliver Morgan, Gabriela Czanner, Mark A. Robinson

**Affiliations:** 1School of Computer Science and Mathematics, Liverpool John Moores University, Liverpool L3 3AF, UK; brian.reilly31@gmail.com (B.R.); G.Czanner@ljmu.ac.uk (G.C.); 2School of Sport and Exercise Sciences, Liverpool John Moores University, Liverpool L3 3AF, UK; O.J.Morgan@2017.ljmu.ac.uk; 3The Celtic Football Club, Celtic Park, Glasgow G40 3RE, UK

**Keywords:** change of direction movements, Global Positioning System, accelerometer, gyroscope, machine learning, random forest, logistic model tree

## Abstract

Changes of direction (COD) are an important aspect of soccer match play. Understanding the physiological and biomechanical demands on players in games allows sports scientists to effectively train and rehabilitate soccer players. COD are conventionally recorded using manually annotated time-motion video analysis which is highly time consuming, so more time-efficient approaches are required. The aim was to develop an automated classification model based on multi-sensor player tracking device data to detect COD > 45°. Video analysis data and individual multi-sensor player tracking data (GPS, accelerometer, gyroscopic) for 23 academy-level soccer players were used. A novel ‘GPS-COD Angle’ variable was developed and used in model training; along with 24 GPS-derived, gyroscope and accelerometer variables. Video annotation was the ground truth indicator of occurrence of COD > 45°. The random forest classifier using the full set of features demonstrated the highest accuracy (AUROC = 0.957, 95% CI = 0.956–0.958, Sensitivity = 0.941, Specificity = 0.772. To balance sensitivity and specificity, model parameters were optimised resulting in a value of 0.889 for both metrics. Similarly high levels of accuracy were observed for random forest models trained using a reduced set of features, accelerometer-derived variables only, and gyroscope-derived variables only. These results point to the potential effectiveness of the novel methodology implemented in automatically identifying COD in soccer players.

## 1. Introduction

Sport scientists and practitioners working in soccer need to understand the physiological demands of soccer match play. This helps to tailor training and injury rehabilitation. Sport-specific training has been demonstrated to yield performance benefits across a multitude of sports, including soccer [1]. Focusing rehabilitation exercises on sport-specific movements and loads has been outlined as a key contributing factor to the successful return to play of athletes that have sustained hamstring [2], anterior cruciate ligament [3] and upper body injuries [4]. To characterise match play, data are obtained from wearable tracking devices which record Global Positioning System (GPS), accelerometer, and gyroscopic data, metrics such as distance covered, number of accelerations, maximum speed and time spent in defined intensity thresholds. Metrics from various technology providers have been used to establish norms in soccer [5], yet the algorithms underlying these metrics are not always known.

This paper focuses specifically on changes of direction (COD) in soccer. COD in soccer refer to instances in a game where a player changes their direction path relative to the path previously travelled [6,7]. Through manual notation of soccer matches, soccer players have been observed to complete >700 turns and swerves in a single game [8], making COD a useful proxy in determining a player’s loading profile [9]. Manual notation, which involves a human expert watching games and extracting COD instances, has been the conventional technique used to record COD [6,8,10,11,12], but this is very time-consuming and labour-intensive and can be subject to reliability issues [13]. One solution to this manual and subjective video annotation process is to develop an automated computerised approach using the multi-sensor wearable tracking devices worn by players. Whilst the manufacturers of these wearable tracking devices may provide COD metrics, the validity, reliability and definition of these variables are unknown. Many studies have attempted to investigate COD using such devices [14], but they have (i) not been attempting to classify COD, (ii) not used actual match play data, and/or (iii) not validated the COD identification against gold standard manual notation analysis. A recent review [14] examining the identification of COD tasks specifically commented that whilst validity and reliability of automated approaches showed promise, future studies should be on-field, with large numbers of COD and standardised COD definitions. This study addresses all of these limitations.

Classification algorithms have been used in similar contexts and studies have shown that machine learning methods may be applied to data obtained from wearable tracking devices in order to develop physical activity classifiers. Specific team sport activities (stationary, walking, jogging, running, counter-movement jumping, tackling) have been classified previously, finding that the logistic model tree was the highest performing model with accuracy of 79–92% [15]. The efficacy of GPS, accelerometer and gyroscopic data to be used as complex physical activity classifiers has also been demonstrated [16,17,18]. These results suggest that there is a potential to use wearable tracking devices to classify team sport activities from soccer match-play data. Quantifying changes of direction from inertial sensor data therefore appears possible.

The aim of this study was to develop a binary automated classification model which can be used to assess player tracking device data and determine the total number of COD > 45° that a player makes in a game, and at what time in the game the COD occurs. The research hypothesis for this investigation is that player tracking device data may be used to classify COD by soccer players with an overall accuracy above 90%. The primary outcome measures are model classification accuracy area under the receiver operating curve (AUC) as well as sensitivity and specificity.

## 2. Materials and Methods

### 2.1. Data Sources

Twenty-three Premier League academy soccer players participated in ten competitive matches (U18s Youth League, U23s Youth League, U18s FA Youth Cup and the Premier League International Cup) during the 2017–2018 and 2018–2019 seasons. Individual player movements were tracked in two ways throughout the game, (1) via traditional 2D digital video capture (Canon XM2, Amstelveen, The Netherlands) mounted on stationary tripods, and (2) with an individual tracking device (Apex Units, StatsSport, Newry, Co. Down, Northern Ireland, UK).

Video footage was transferred to a laptop computer and changes of direction were analysed using performance analysis software (Sportscode Gamebreaker Plus 10.3.36, Sportscode, NSW, Australia). A flow chart decision tree was created to identify and characterise a COD (Figure 1), where a COD was defined as a path change caused by an identifiable plant of a leg. A path change was described as a change in path travelled relative to the path previously travelled by the player [6]. COD were discarded (not recorded) if the motion occurred immediately post a walk, if the player performed an arched run immediately after the motion, or if the player was in possession of the ball. This COD identification system, including strong intra-rater reliability and comprehensive descriptions of the COD from these matches is reported elsewhere [12]. Upon completion of video analysis, the data contained three columns; ‘Start Time’ and ‘End Time’ which signified, respectively, the beginning and end of an observed COD event, and ‘Category’, which characterised the COD angle as one of ‘0–45 DEG’, ‘45–90 DEG’, ‘90–135 DEG’ and ‘135–180 DEG’. The average duration between the ‘Start Time’ and ‘End Time’ for the observed COD was 3.1 s.

Tracking devices were worn by all players in specially designed vest-pockets situated on the players’ backs between their shoulder blades, monitoring players’ movements throughout the course of the games (Figure 2). Tracking device data were exported as a raw text output containing all sensor outputs, and was then used for analysis. ‘Latitude’, ‘Longitude’, ‘Speed’ and ‘Instantaneous Acceleration Impulse’ (IAI) were sampled at a rate of 10 Hz using GPS technology. Accelerometer data were sampled at 100 Hz to record accelerations and decelerations in the X, Y and Z planes. Gyroscopic data were also sampled at 100 Hz in the X, Y and Z planes to capture the rotations of the tracking device unit.

### 2.2. Data Processing and Exploratory Analysis

The GPS data were divided into 2.5 s intervals for data processing and feature extraction purposes. Two such divisions were implemented, one starting at 0 s and another at 1.25 s, to allow for overlap, for each player’s data. The selection of an interval duration of 2.5 s was made in consideration of the average COD event duration recorded in the video analysis data (3.1 s), and the aim of recording minimal noise in the data that could potentially ambiguate the COD. The average number of 2.5 s segments in each game-half of soccer was 1140 +/− 87 (±standard deviation), per player. This 2.5 s interval duration has significant implications for the data processing stage. For each 2.5 s interval, a mean, maximum and minimum value was obtained for the ‘Speed’ and IAI variables, as well as each of the accelerometer and gyroscopic variables in the X, Y and Z planes.

The GPS co-ordinates were used to create a novel variable, named ‘GPS-COD Angle’, which aimed to identify the change in running angle (if any) that occurred over the course of the 2.5 s interval. As the GPS data were sampled at a rate of 10 Hz, each 2.5 s interval contained 25 sets of co-ordinates representing the players’ location on the field. Average values for the ‘Latitude’ and ‘Longitude’ variables were obtained for the first 0.5 s (sets of co-ordinates 1–5), the median 0.5 s (sets of co-ordinates 11–15) and the final 0.5 s (sets of co-ordinates 21–25) for each 2.5 s segment. The geo-spatial location of these 3 values were then found and the distance between each point recorded, mapping out a signature triangular shape for each 2.5 s segment (Figure 3). The cosine rule was then used to determine the angle at the second point of the triangle. This value subtracted from 180 was used to produce a GPS-COD Angle variable for each 2.5 s segment. The cosine rule formula used to calculate the GPS-COD Angle variable (Equation (1)):(1)$$‘\mathrm{GPS}-\mathrm{COD}\;\mathrm{Angle}’=180-\left[cos^{-1}\left(\frac{a^{2}+c^{2}-b^{2}}{2ac}\right)\right]$$
where; *a* = Distance between 2nd and 3rd points in triangle, *b* = Distance between 1st and 3rd points in triangle and *c* = Distance between 1st and 2nd points in triangle.

The ground truth is the manually annotated video analysis data where the annotator indicated when a COD > 45° occurred. This was coded as a binary variable with 1 or 0, for presence or absence of a COD > 45°, respectively. This was carried out by segmenting the video analysis data in to 2.5 s intervals, as previously with the tracking data, and assigning a value of 1 if the segment contained a COD > 45°, otherwise assigning a value 0. This newly created outcome variable acted as the gold standard or ground truth which was used to train and test the machine learning methods.

This data pre-processing procedure was carried out for each game-half of player data to produce 25 potential predictive variables, plus the outcome ground truth variable. Of the 46 game-halves available for data processing, 14 were omitted for various reasons. Reasons for omission included the presence of faulty sensor data (e.g., the presence of unrealistic GPS co-ordinates likely due to insufficient satellite coverage) or a lack of clarity surrounding temporal alignment. In certain instances, the exact start time of the game-half was unclear and could not be determined from the tracking device data as players tracking devices are powered on for several minutes before the referee blows the whistle to begin gameplay. The video footage data were used in certain instances as a tertiary data source to add some clarity in this regard. The final analysable data set included, therefore, 32 halves of player data, from 19 different players.

These data were combined to produce approximately 68,000 rows of data (2.5 s intervals), with ~58,000 containing an outcome variable of 0 (‘no COD’) and ~10,000 containing an outcome variable 1 (COD > 45° occurred/‘yes COD’). Exploratory analysis was carried out to obtain an initial understanding of associations. Independent samples *t*-tests were conducted to compare the means of each of the 25 variables according to outcome group.

### 2.3. Classification Model Development and Validation

Classification models were created using one statistical and one machine learning method, logistic model tree (LMT) and random forest (RF). These methods were selected due to their demonstrated ability to produce successful physical activity classification models in the past [15,19]. An imbalance existed in the data: ~58,000 data entries with ground truth 0 (‘no COD’), compared to only ~10,000 data entries with ground truth 1 (‘yes COD’). The class imbalance can limit the effectiveness of machine learning methods [20]. Hence, to address this issue, the larger class (‘no COD’) was down-sampled so that each class was of equal size (~10,000 rows).

To achieve a fair understanding of the generalisability of the classifiers, 5-fold cross-validation was used to evaluate the performance of the models, repeated 10 times. AUC, sensitivity and specificity were the key metrics used to examine the effectiveness of the various models created.

A variety of hyperparameters were tuned to optimise LMT model performance. The numeric significance level (Alpha), which determines if a node is split according to the *p*-value for any parameter stability test, was set to 0.05. The minimum size set for the number of observations in a node was fixed at 50. The maximum depth of the tree model produced was set to 4. Post-pruning was carried out in accordance with the minimisation of Akaike information criterion (AIC).

Investigation of the importance of the 25 potential predictors was carried out. Stepwise procedure was used to determine a reduced set of predictive variables which was then used to train and evaluate an RF model. RF models were also trained and tested exclusively using variables derived from the collected and processed accelerometer data, and exclusively using the gyroscopic data. RF hyperparameters ‘mtry’, which is equal to the number of randomly selected predictors at each tree node, and ‘number of trees’ were tuned to produce the highest performing model.

Finally, the choice of the classification acceptance threshold was investigated. The significance of this parameter is that it defines the identified posterior probability that would cause the model to classify an instance as ‘yes COD’ or ‘no COD’. When the classification model is evaluated with test data, it ultimately outputs a posterior probability value (between 0 and 1) that represents the model’s objective belief as to whether a COD has occurred or not within a time segment, given the observed values of the predictor variables for that segment. An optimal classification acceptance threshold was chosen in consultation with sport science practitioners, so that it achieves equality of sensitivity and specificity.

## 3. Results

In the study data, 23 potential predictors showed a difference between the two groups: ‘yes COD’ (n = 10,188) and ‘no COD’ (n = 58,040), statistical significance level of 0.01 (Table 1). The two variables that were not statistically associated with the outcome were ‘Mean Gyro.Y’ (*p* = 0.3) and ‘Mean Gyro.Z’ (*p* = 0.38). Across the 23 statistically significant metrics, the largest differences were seen for (in terms of the ‘yes COD’ group, relative to ‘no COD’ group): GPS-COD Angle (the new variable) increased by 25%, Mean Speed increased by 44%, Min Speed increased by 33%, Max Speed increased by 47%, Max Accel X increased by 80%, Max Accel Y increased by 49%, Max Accel Z increased by 57%, Max Gyro X increased by 63%, Max Gyro Y increased by 64%, Max Gyro Z increased by 62%.

A visual inspection of the difference in the distribution of values for the GPS-COD Angle variable between outcome groups showed substantial overlap (Figure 4). While there is a lot of overlap between outcome groups, the observed GPS-COD Angle differed in terms of mean (‘no COD’ = 27.06, ‘yes COD’ = 33.03, *p*-value = < 0.01), median (‘no COD’ = 13.73, ‘yes COD’ = 20.73) and interquartile range (‘no COD’ = 5.44–32.77, ‘yes COD’ = 8.91–45.39) (Table 1). The observed overlap points towards the need to include other potential predicting variables in the classification of COD > 45°.

Stepwise regression led to the selection of 16 variables for training the RF model with a reduced set of features, which were: GPS-COD Angle, Mean Speed, Mean IAI, Mean Accel Y, Mean Accel Z, Mean Gyro X, Mean Gyro Y, Mean Gyro Z, Min. Speed, Min. Accel Y, Min. Gyro Y, Min. Gyro Z, Max. IAI, Max. Accel Y, Max Accel Z and Max. Gyro Z.

The highest performing model for the detection of COD > 45° is the RF model with the full set of variables (Table 2). It has the highest mean AUC, Sensitivity and Specificity. The RF models can be characterised by their significantly higher performance levels compared to the LMT model, at a level of significance of 0.05. This is seen by the fact that the 95% confidence interval of the LMT model does not intersect with any of the RF models. Furthermore, it can be seen that the prediction results of the various RF models are not statistically different from each other, with closely matched levels of ability as it pertains to correctly identifying COD. This is seen from the overlap of their 95% confidence intervals, while the similarity in the lower and upper bounds also suggests the same precision.

The variables which had the most explanatory power after training the RF model containing the full set of features were investigated (Figure 5). The novel GPS-COD Angle variable was deemed to be the most important variable in predicting COD (variable importance = 100), followed by ‘Mean_AcclX’ (92.3), and so on.

The effect that the acceptance threshold (decision threshold) has on the true positive rate (TPR, also called sensitivity) and true negative rate (TNR, also called specificity) of the best performing RF model was examined (Figure 6). As expected, as we increase the threshold, the sensitivity (TPR) of detecting the COD > 45° increases, while the specificity (TNR) of correctly detecting no COD > 45° decreases. The decision threshold = 0.62 is equivalent to the point at which TPR and TNR are equal.

The complete confusion matrix produced by five-fold 10 repeated cross validation (n = 611,280) is shown (Table 3). With a default model classification acceptance threshold (0.5), sensitivity is 0.941, with specificity far lower, at 0.772. The balancing effect of the adjustment of the acceptance threshold to 0.62 is shown to produce an equal level of model specificity and sensitivity (0.889).

## 4. Discussion

The aim of the present study was to investigate if it was possible to develop a binary automated classification model from player tracking device data to determine whether a player completed a COD > 45°. The first principal finding is that the novel GPS-COD Angle variable, when used in random forest classifier, showed the potential to automatically identify COD as they occur throughout the game with AUC +/− standard error of 0.957 +/− 0.01. The implications of this finding are multifaceted, both in terms of the specific method-related factors that led to its occurrence and the impact on the classification of soccer player movements.

The second principal finding is that the number of predictors (features) can be reduced. The RF models trained using the full set of features and a reduced set of 16 selected features using stepwise regression produced AUC scores of 0.957 +/− 0.01 and 0.954 +/− 0.01, respectively (Table 2). Furthermore, RF model accuracy is comparable when using accelerometer or gyroscope data only. The powerful performance of RF is consistent with the findings of a study that used tracking device data (GPS and accelerometer) to detect real-life physical activity types [19], where one general RF model was able to classify a variety of tasks with a classification accuracy of 84% when assessing unseen test data. The powerful performance of RF as it pertains to this particular data set is underlined when it is compared to that of the LMT model (Figure 6). In validation tests of the LMT model, a far inferior AUC of 0.747 +/− 0.02 was observed. These results contrast with the findings of a previous movement classification study, whereby LMT was shown to outperform RF in terms of the development of a physical activity classifier [15]. The success of RF in relation to the present data set may be attributed to its cited abilities with respect to handling highly non-linearly correlated data, tuning simplicity and robustness to noise in the data [21].

The comparison of the distribution of values for each individual variable between outcome groups (Table 1), may also help to explain the powerful nature of the classification models produced. The means for all but two variables were found to be significantly different. This distinction between the two classification groups increases the potential power of machine learning methods. The observed difference in the two groups also suggests that the 2.5 s segment duration used is an appropriate interval in which a COD may be captured.

Multiple aspects of the results of the present study point to the effectiveness of the novel trigonometry-based approach used to obtain the GPS-COD Angle variable. The significantly higher mean value for the ‘yes COD’ group compared to the ‘no COD’ group (Table 1, Figure 4) suggests that the applied method was at least somewhat successful in measuring the change in movement angle of players at any point in the game. Furthermore, the GPS-COD Angle variable ranks highest in terms of variable importance in the most powerful RF model (Figure 5). Variable importance provides an insight into which variables in the data set provide the model with the most explanatory power, and as such, this result serves to underline the validity of the GPS-COD Angle variable.

Accelerometer-generated variables were found to have the same COD predictive power as those generated using gyroscope technology, in terms of RF AUC, sensitivity and specificity (Table 2). The accelerometer predictive power is higher by 0.5% only, and this was not significant. The high predictive power associated with the accelerometer data is unsurprising, given the prevalent use of accelerometer data in previous physical activity classification studies [15,19]. However, it should be noted that all RF models offered statistically equal performance; practically differing by 1% at most, in accuracy. Accelerometer and gyroscopic sensors embedded in tracking devices are both equally important tools to be used in the classification of COD > 45°.

With the default classification acceptance threshold of 0.5, the RF models in general exhibit high levels of sensitivity and significantly lower levels of specificity. The best performing RF model, trained on the full set of features, has a sensitivity of 0.941, and a specificity of 0.772 (Table 1). In practice, this implies that the model would be successful in identifying COD 94.1% of the time, which is a promising mark. However, when the associated error rate of 0.19 in predicting instances where COD do not occur is projected on to a game-half of soccer of average length (1140 2.5 s segments) that contains the average number of segments containing COD (199), it would result in the reporting of 377 COD. In effect, this means that an 89.4% over-reporting of COD would take place. To somewhat address this weakness of the model, the classification acceptance threshold was set at 0.62, balancing sensitivity and specificity at 0.889 (Table 3, Figure 6). This has the effect of reducing the number of false positives recorded, while maintaining a relatively low number of false negatives. Within the context of an average game-half of soccer, this would, on average, result in the identification of 177/199 2.5 s segments containing COD and the incorrect classification of 104/941 2.5 s segments that do not contain a COD, as indeed containing one. Choosing the level at which the classification acceptance threshold is set may require further thought and investigation, or else it may be set at the discretion of practitioners who decide the relative importance of sensitivity and specificity in varying situational circumstance. Furthermore, using additional sensors in alternative locations e.g., the foot [22], may provide unique data to the trunk-mounted sensor and further improve COD classification.

This associated error rate represents the primary weakness of the model in practice. Another area of concern with the methods of the present study relates to the density of COD. This refers to the frequency at which COD occur. For the present study, it was assumed that only one COD can take place in each 2.5 s segment, however, anecdotally it is known that a player may complete more than one COD within this time frame. This may therefore limit the effectiveness of the model in a real-world setting. No published literature was found dealing with this issue of COD density. A deeper knowledge of this area could help to inform the development of a more effective COD classification model. Furthermore, in the data processing stage, a segment that overlapped in any way with the occurrence of a COD was classified as ‘yes COD’. This means that, in theory, even if the majority of a COD recorded during video analysis had occurred outside of a given 2.5 s segment, but there was a slight overlap, the segment would still be classed as ‘yes COD’. This could potentially have had a negative impact on model training.

## 5. Conclusions

This study serves to fill an identified gap in the literature in the area of COD in soccer [14], being the first to attempt to automate the classification of COD and to use actual match play data, while also validating the COD identification against gold-standard manual notation analysis data.

In this study, a transparent COD > 45° classification model using player tracking device data was developed. With some adjustments, the novel approach that was developed and applied in the present study has shown the potential to be effective. In its current form, there is scope for practitioners to use the model and treat the output as a proxy for a player’s COD count, in the knowledge of the associated error rates. In future, studies that follow the framework of the present investigation may consider the trialling of a range of different movement capture durations (e.g.,: 1 s, 1.5 s, 2 s) and the issue related to COD density may be addressed to produce a model with a reduced error rate, whicould effectively be used and trusted to gather accurate information about in-game COD. The present study has set the stage for further development and improvement of game-based automated detection of COD.

## Figures and Tables

**Figure 1 sensors-21-04625-f001:**
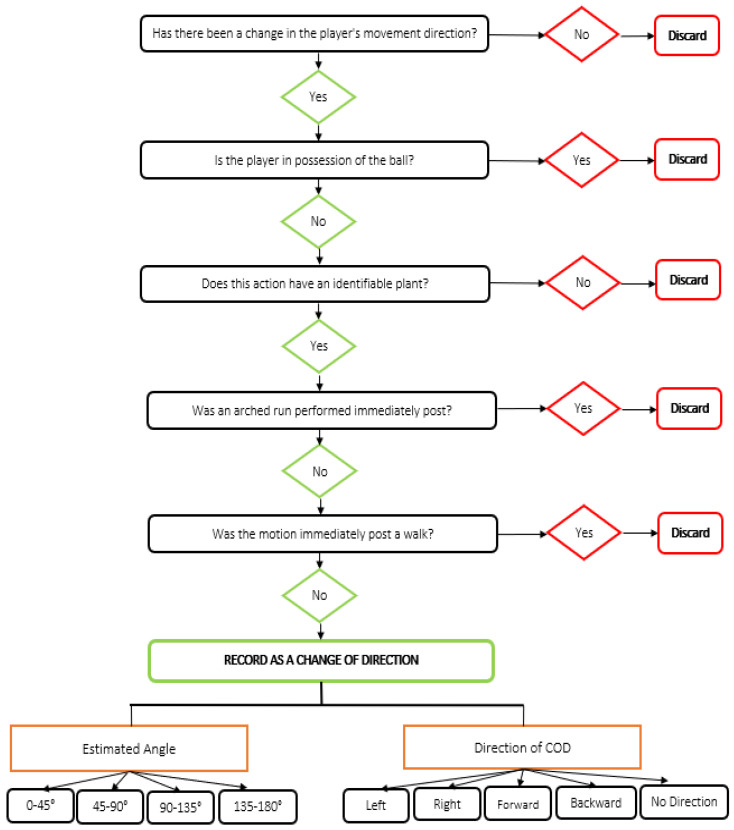
COD decision tree used for video analysis.

**Figure 2 sensors-21-04625-f002:**
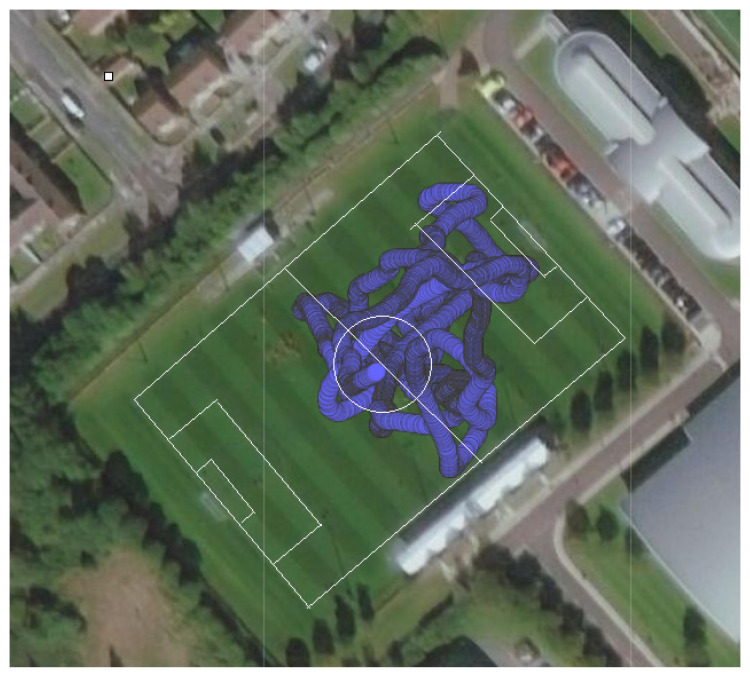
GPS-generated data showing a player’s movements over the course of the first 8 min of a game, with pitch markings superimposed for reference.

**Figure 3 sensors-21-04625-f003:**
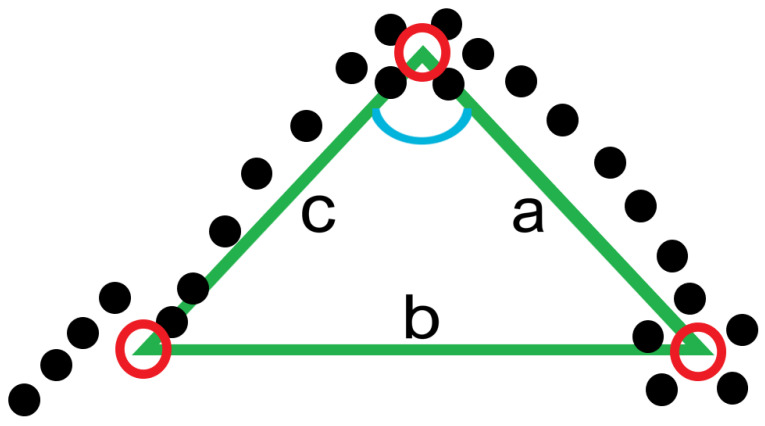
Illustration of the method used to determine the ‘GPS-COD Angle’ variable for a single 2.5 s segment. 25 black dots represent full player movement. Red circles represent average location of starting 5, median 5 and end 5 points. Blue angle = angle of interest.

**Figure 4 sensors-21-04625-f004:**
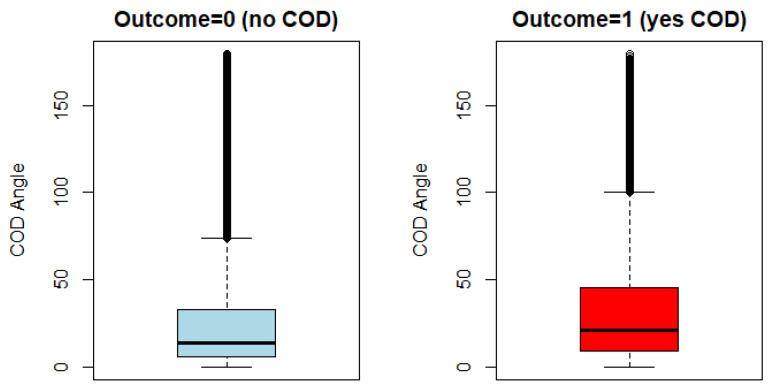
Box-plot comparison of the novel GPS-COD Angle values for observations from categories ‘no COD’ and ‘yes COD’.

**Figure 5 sensors-21-04625-f005:**
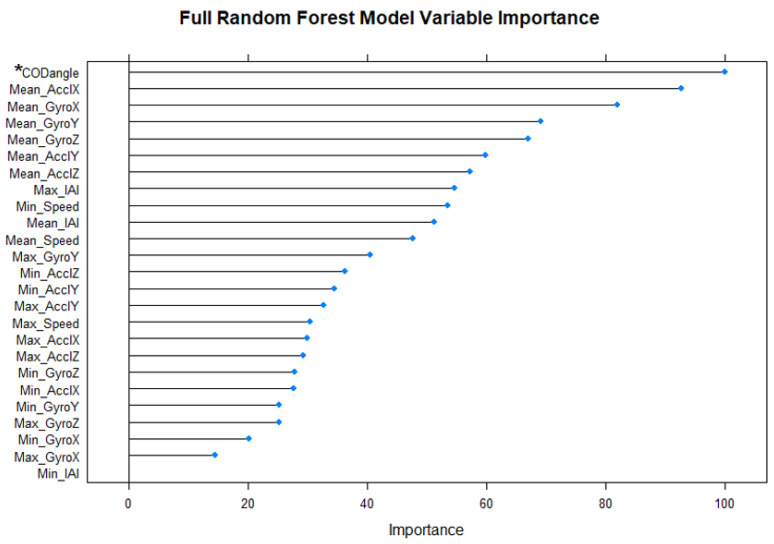
Variable importance for all 25 potential predictors (features) in RF model. * Note: ‘CODangle’ = ‘GPS-COD Angle’ variable.

**Figure 6 sensors-21-04625-f006:**
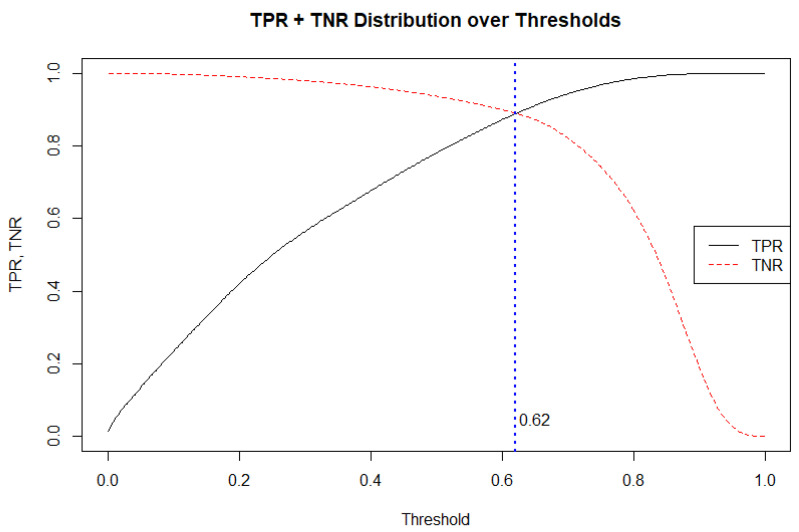
True positive rate (TPR, i.e., sensitivity) and true negative rate (TNR, i.e., specificity) of RF model using full set of features, plotted over a full range of acceptance thresholds.

**Table 1 sensors-21-04625-t001:** The 25 potential predictors for COD > 45°. The mean, median and interquartile-range values for variables of each outcome class (prior to down-sampling). *p*-value derived from independent samples *t*-test.

	Outcome = 0 (n = 58,040) ‘No COD’			Outcome = 1 (n = 10,188) ‘Yes COD’			
Variable	Mean	Median	Interquartile Range	Mean	Median	Interquartile Range	*p*-Value
GPS-COD Angle	27.06	13.73	5.44 to 32.77	33.73	20.73	8.91 to 45.39	<0.001
Mean Speed	1.71	1.37	0.85 to 2.39	2.47	2.34	1.58 to 3.16	<0.001
Mean IAI	1.46	1.44	1.06 to 1.86	1.85	1.85	1.56 to 2.16	<0.001
Mean Accel.X	−0.012	−0.013	−0.070 to 0.046	−0.008	−0.011	−0.080 to 0.063	<0.001
Mean Accel.Y	0.89	0.93	0.86 to 0.98	0.85	0.88	0.78 to 0.96	<0.001
Mean Accel.Z	0.54	0.52	0.45 to 0.61	0.59	0.58	0.49 to 0.68	<0.001
Mean Gyro.X	4.83	6.55	−36.86 to 47.72	13.03	8.4	52.84 to 78.03	<0.001
Mean Gyro.Y	28.55	25.4	−102.86 to 160.15	24.73	22.73	−177.56 to 234.91	0.3
Mean Gyro.Z	36.9	36.25	−59.39 to 137.83	33.98	30.72	−143.56 to 213.61	0.38
Min. Speed	1.04	0.83	0.3 to 1.41	1.38	1.19	0.59 to 1.98	<0.001
Min. IAI	0.043	0	0 to 0.06	0.034	0	0 to 0.03	<0.001
Min. Accel.X	−0.99	−0.65	−1.31 to −0.38	−1.75	−1.46	−2.24 to −0.94	<0.001
Min. Accel.Y	−0.25	−0.29	−0.79 to 0.56	−1.03	−0.85	−1.34 to −0.59	<0.001
Min. Accel.Z	−0.16	−0.04	−0.39 to 0.23	−0.62	−0.44	−0.76 to −0.25	<0.001
Min. Gyro.X	−1984.3	−1280	−2764.8 to −563.2	−3465	−2918.4	−4505.6 to −1894.4	<0.001
Min. Gyro.Y	−2064.9	−1536	−2816 to −716.8	−3404.9	−2918.3	−4249.6 to −1996.8	<0.001
Min. Gyro.Z	−1378.9	−1075.2	−1894.4 to −563.2	−2298.8	−2048	−2867.2 to 1433.6	<0.001
Max. Speed	2.38	1.88	1.26 to 3.33	3.51	3.4	2.54 to 4.35	<0.001
Max. IAI	4.22	4.42	3.39 to 5.31	5.01	5.25	4.58 to 5.64	<0.001
Max. Accel.X	0.97	0.61	0.34 to 1.29	1.75	1.41	0.88 to 2.26	<0.001
Max. Accel.Y	3.24	2.4	1.64 to 4.49	4.84	4.6	3.36 to 5.98	<0.001
Max. Accel.Z	2.04	1.54	0.96 to 2.71	3.2	2.87	2.09 to 3.89	<0.001
Max. Gyro.X	1687.5	1177.6	614.4 to 2252.8	2746	2355.2	2536 to 4532	<0.001
Max. Gyro.Y	2158.3	1638.4	768 to 2918.4	3529.2	3020.8	2048 to 4532	<0.001
Max. Gyro.Z	1451.4	1177.6	665.6 to 1945.6	2349.7	2099.2	1536 to 2918.4	<0.001

**Table 2 sensors-21-04625-t002:** Prediction results of five models. The AUC mean, 95% confidence intervals, mean sensitivity and specificity of each model were calculated using five-fold 10 repeated cross-validation. The RF models differ in the set of potential predicting variables (features) used.

Model	AUC Mean	AUC Mean 95% Confidence Interval	Sensitivity Mean	Specificity Mean
Logistic Model Tree (All features)	0.747	0.745–0.749	0.716	0.672
Random Forest (All features)	0.957	0.956–0.958	0.941	0.772
Random Forest (Reduced set of features)	0.954	0.953–0.955	0.939	0.767
Random Forest (Accelerometer variables)	0.953	0.952–0.954	0.936	0.759
Random Forest (Gyroscope variables)	0.948	0.947–0.949	0.933	0.752

**Table 3 sensors-21-04625-t003:** Confusion matrix for RF model using full set of features after five-fold 10 repeated cross validation. Classification acceptance threshold set at 0.5 (left) and 0.62 (right).

Classification Acceptance Threshold = 0.5 (Default)					Classification Acceptance Threshold = 0.62 (Adjusted)				
n = 611,280	Actual: no COD	Actual: COD	Error Rate	Sens./Spec.	n = 611,280	Actual: no COD	Actual: COD	Error Rate	Sens./Spec.
Predicted: no COD	235,793	18,013	0.07	Sensitivity = 0.941	Predicted: no COD	271,845	33,867	0.11	Sensitivity = 0.889
Predicted: COD	69,667	287,627	0.19	Specificity = 0.772	Predicted: COD	33,795	271,773	0.11	Specificity = 0.889

## Data Availability

The data presented in this study are available on request from the corresponding author.

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
