# Peer review of "Automated Classification of Changes of Direction in Soccer Using Inertial Measurement Units"

_sensors, 2021, doi:10.3390/s21144625_

Round 1
Reviewer 1 Report
The manuscript, Automated Classification of Changes of Direction using Inertial Measurement Units During Soccer Match Play. The methods are explained with consistent tests on experimental data sources. The paper is well structrured and clearly presented.
Nevertheless, I have some suggestions for further improvements:
1 - The authors should better put their results in relation to what has been achieved in literature.
2- The article lacks a more complete review of the state of the art. The problem presented has certainly been addressed in the existing literature, and the reviewer wonders whether or not previous contributions have considered the same approach taken in the current paper. This is not clear enough in the current text, even though this is a case study. The manuscript needs a thorough revision in this respect.
3- Authors should reconsider what is the main novelty of their work and why it is relevant to this community. This should be made clear in the manuscript.
4 - Apart from the application, and the interesting study presented, I have the feeling that only well-known machine learning methodologies are used.
I would like the authors to comment on this.
5 - The authors should reconsider what is the main novelty of their work and why it is relevant to this community.
6 - Figures 1 and 2 should be cited in the text.
7 - The COD decision tree presented in Figure 2 needs to be better explained in the text.
8 - The equation presented on page 4 line 132, should be numbered and properly cited in the text.
9 - The quality of Figures 2. Idem Figure 3
10 - What is the reason for the classification models being created using a Logistic Model Tree (LMT) and Random Forest (RF)? This is not clear in the text.
12 - What is the criteria for choosing the 25 potential predictors? Are all of them used in the classification model?
13 - Is any dimensionality reduction process proposed?
Reviewer 2 Report
General Comments
The goal of this investigation was to develop an automated classification model for COD>45° in soccer matches based on GPS or IMU data to replace the time-intensive gold standard of manually viewing and tagging 2D video. The author showed promise in their methods and have provided recommendations for future improvements. The work is novel and important for assessment of player movement. Overall, the manuscript is well written and easy to follow. I only have a couple of specific comments.
Specific Comments
Abstract
None
Introduction
None
Methods
- No reference from text to Figure 1.
- Line 94, The range of COD angles does not match the angle categories of Figure 2. Shouldn’t they be the same?
- Line 113, if the average COD was 3.1s why was the interval duration 2.5s? Seems like this needs a little further explanation.
Results
- Figure 5 caption, line 251, I don’t see ‘COD Variable’ anywhere in the figure, do you mean ‘CODangle’?
Discussion/Conclusions
None
Round 2
Reviewer 1 Report
Considering the revised version of the manuscript, I recommend acceptance in its present form with no revisions.